# Case Study: 3D Modelling and Printing of a Plastic Respirator in Laboratory Conditions

Miriam Pekarcikova [1,*], Peter Trebuna [1], Marek Kliment [1] and Stefan Kral [2]

1   Department of Management, Industrial and Digital Engineering, Faculty of Mechanical Engineering, Technical University of Košice, 040 01 Košice, Slovakia; peter.trebuna@tuke.sk (P.T.); marek.kliment@tuke.sk (M.K.)
2   Slovak Legal Metrology n.o., 974 01 Banská Bystrica, Slovakia; kral@slm.sk
*   Correspondence: miriam.pekarcikova@tuke.sk

**Abstract:** The importance of 3D printing is primarily that it enables customized production and, through Industry 4.0 technology, enables decentralized production. The article deals with the issue of 3D modelling and 3D printing of plastic respirators in the laboratory conditions of the authors' workplace. In the above case study, the process of creating 3D models of individual parts of a plastic respirator and the production of a given model using a 3D printer is processed. The article also outlines the trends in 3D printing in connection with Blockchain and their importance on the Supply Chain.

**Keywords:** 3D printing; 3D modelling; CAD software; respirator

## 1. Introduction

3D printing can be understood as an additive production procedure, i.e., a process during which a model is produced from electronic data in physical form by the gradual layering of material. The most frequently used materials include gypsum composite, various types of plastics, waxes, resins, metals, glass, but also ceramics. The most common ways to create an electronic model are CAD modelling and 3D scanning. Due to inaccuracies, the scan must be fine-tuned in a graphics program after 3D scanning. This also includes technical data used in medical imaging systems. The model that is created in the computer program must be divided into thin horizontal layers and the model modified in this way is ready to be sent as information to a 3D printer. Using the latest technologies, a 3D printer can apply the solid material layer-by-layer [1–4]. Subsequently, the layers gradually solidify until they are joined together to form the final product. With 3D printing, it is possible to produce parts as needed and reduce inventory. According to general studies, companies can save up to 85% of their on-demand shipping costs when producing components on-site using 3D printers. The potential for 3D printed spare parts is huge. According to the DHL report [5,6], the share of unused parts or overcapacity in companies is more than 20%. Many car manufacturers have to store spare parts for each vehicle model for 7 to 10 years. Thanks to 3D printing, the supply chain is expected to reduce costs by 50 to 90% for slow-moving spare parts. Unlike conventional production methods, 3D printing produces almost no waste. Conventional production methods will be replaced by 4D printing, which adds a fourth dimension to the components—functionality [7–9]. The simulation experiments were done on the basis of changing of the input parameters [9,10].

In connection with 3D printing, it is possible to highlight the advantages of Blockchain, which can be used effectively within the Supply Chain, especially in guaranteeing relevant information about material, production, quality, chain entities, etc. [11–14]:

- Transparency: Blockchain can require confirmation from all parties;
- Traceability: Blockchain can provide the status of modular housing products with a timestamp;

- Immutability: Blockchain can offer a tamper-proof solution;
- Decentralization: Blockchain can prevent them entirely controlled by one party;
- Privacy-preserve: Blockchain can encrypt them by using hashing algorithms;
- Smartness: smart contracts facilitate the automatic execution process in transparency, traceability, immutability, decentralization, and privacy.

The fusion of 3D printing and Blockchain technologies will enable fast forensic analysis in the event of a part failure. Blockchain allows producing components, resp. use local 3D printers to make parts. Blockchain smart contracts and hashing can be used very effectively for data and process integrity. The benefits include the following [11–14]:

- Delivery times are shortened by creating a production scheme with a global network of 3D printers;
- Back-analysis of parts and reduction of pressure on OEM costs;
- Cost reduction for value-added activities;
- More flexible inventory management through digital assets;
- Elimination of transport costs by using digital transfer via the cloud to the place of use;
- Elimination of storage costs, as parts will be stored in a digital warehouse;
- Elimination of duties and charges as objects move digitally across borders;
- Effective form of payment via Bitcoin;
- Setting standards for digital fibres and digital twins;
- Each object will have a certificate of authenticity, which will prevent fake parts from entering supply chains.

A very important company in the aerospace industry in connection with the combination of 3D printing and blockchain is the company VeriTX, which in the source [13,15] presents his own experience in this direction. One of them is the following [13,15]:

> *"When an F-15 Eagle fighter needed a metal part, its commanders turned to VeriTX's virtual marketplace, where they were able to find a nearby supplier who could make it on a 3D printer and deliver it in six hours. Using the old printed catalogue system, it would have taken 265 days, on average, to get the jet flying again."*

The importance of Blockchain is generally in the protection and sharing of data related primarily to print licenses, production process data, material origin information, tests and simulations, payment records, and parts certification [14].

In a study by Klöckner, M., [14] Table 1 contains material that points to the importance of Blockchain in connection with 3D printing. The study defines the potential for business model innovation for local manufacturing, shared factories, and secure design marketplaces well-crafted findings of authors to increase the innovative capacity of business models.



**Table 1.** Blockchain in 3D printing affects all business model elements [14].

| Business Model Opportunities | Value Proposition | Value Creation | Value Capture | Value Network |
|---|---|---|---|---|
| Local manufacturing | • Offer an improved value delivery (local, on-demand printing) <br> • Offer less expensive products (cut logistics costs) | • Secure transfer of design files <br> • Production outsourcing to local manufacturers, service providers, or customers <br> • Full transparency about a printed part's life cycle | • Improved cost structures due to reduced transaction costs <br> • Altered cost structures from fixed costst to variable costs | • 3D printing service providers <br> • Prosumers (higher customer centricity) <br> • All stakeholders sharing part-related data |
| Shared factories | • Offer flexible production capacities <br> • Offer an improved value delivery | • Secure transfer of design files <br> • Secure and efficient blockchain-based payment processes <br> • Counterfeit products become detectable | • Monetization of unused production capacity <br> • Improved cost structure due to access to external capacities (decreasing fixed cost) <br> • And maximization of production utilization | • Firms renting production capacities <br> • Firms offering <br> • 3D printing service providers |
| Secure design marketplaces | • Offer further 3D designs <br> • Offer more customized products | • Secure transfer of design files <br> • Increasing design variety and customization options | • Monetization of unused designs <br> • Reduced development costs through external design procurement | • External designers <br> • New customers |

The study by Klöckner, M., ref. [14] also mentions the pitfalls that currently hinder the use of blockchain by companies in practice. These barriers include the development of stakeholder management concepts, corporate cultures that do not support the sharing and development of ecosystems, and regulatory conditions regarding the protection of personal data. At the technical level, the method of linking the physical printing parts and the digital ledger of the blockchain requires further attention. Large companies operating in the aviation industry, e.g., Honeywell, Air New Zealand, and Moog can launch pilot projects to improve blockchain usability on a wider scale. A Blockchain platform for 3D printing value chain is shown in Figure 1, the authors [14] adapted the scheme from Gibson, et al. [16].

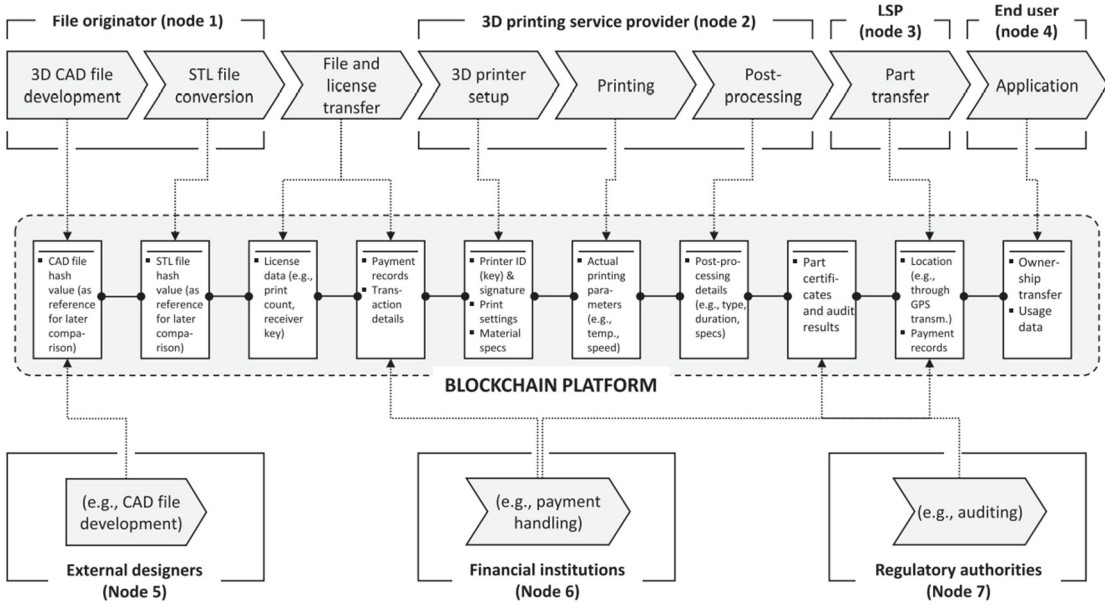

**Figure 1.** Blockchain platform for 3D printing value chain [14].

## 2. Description of the Selected Object for the 3D Printing Process

Drawing CAD software is used to create 3D models for 3D printing as well as regular production [11]. The market offers several to choose from. Each of them is something specific and it is therefore up to the end-user to choose. More advanced CAD software includes Solidworks, Catia, Creo, Inventor, etc., available to the authors' workplace. Creating models is not easy. In this work, Solidworks was used to modify the models (mainly the respirator).

### 2.1. Modelling of a Respirator Plastic Filter

The model was created using basic geometric shapes (prism, pyramid, cylinder). This is, of course, associated with the use of a three-dimensional coordinate system. In Solidworks, it is marked with x, y, z axes. Initially, the basic shape of the grid (see Figure 2) was created only in the two-dimensional coordinate system x, y.

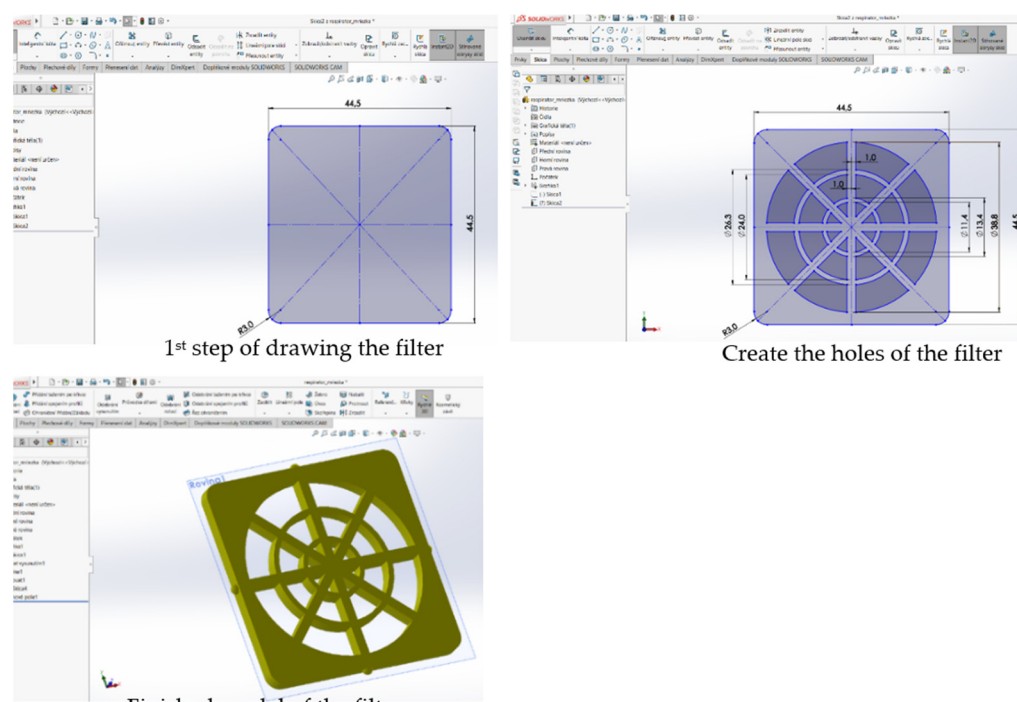

1st step of drawing the filter

Create the holes of the filter

Finished model of the filter

**Figure 2.** Finished model of a 3D plastic filter.

## 2.2. Modelling the Respirator Cap

Again, a sketch was drawn first and then the left side was created by mirroring to create a closed profile (see Figure 3). Then, the sketch was inserted into the front plane and an "offset" was created for it via the "offset entities" example. Again, this is only a 2D model.

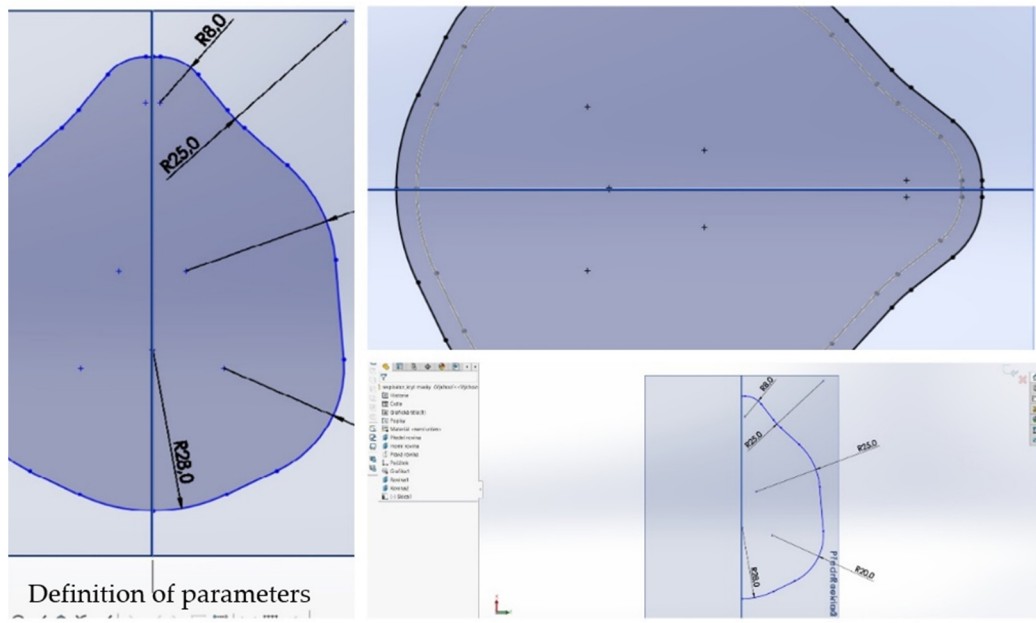

Definition of parameters

**Figure 3.** Drawing a sketch, mirroring it, and creating an "offset".

The mask cover is adjusted to four planes (plane 1 and 2, front plane, upper plane) before joining the profiles. Only then are the profiles joined using the "Join profiles" command. Next, the length of the planes was adjusted. The profiles were connected, and

the model was rotated so that we could see the core. Then the core is marked with the function "shell" (see Figure 4).

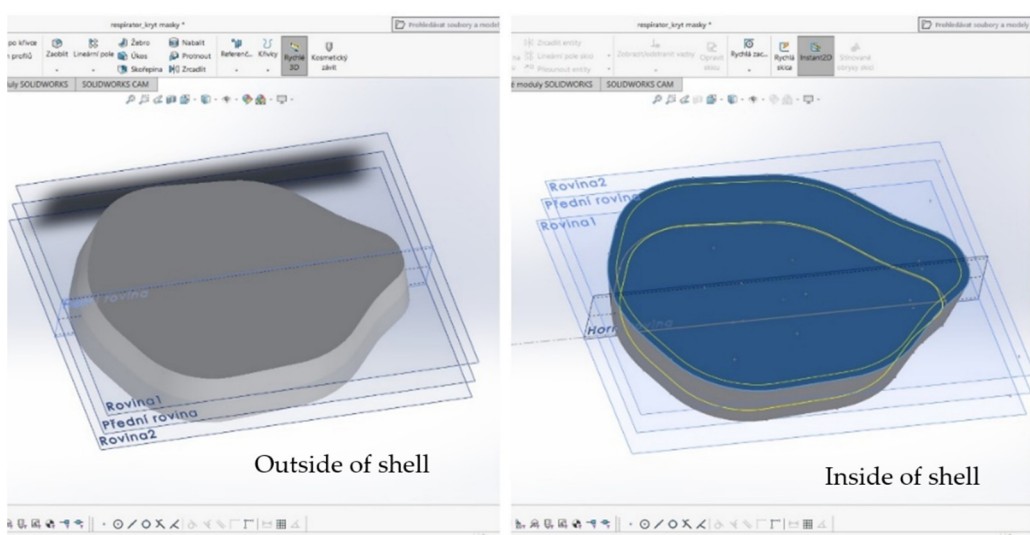

**Figure 4.** Joining profiles and creating a "shell".

To be able to breathe fully through the cover, there must be some openings in the front wall so that air can flow there. To create these holes, we needed to see the model in 2D view. Using straight lines, we laid out one hole, centered in the center. Its length was 36.7 mm and width 5 mm. Then another four were created on the left side of the model with a spacing of 4 mm. The mirroring process was applied to the right side, the alignment was checked, and the cut-out applied.

For the protrusions formed on the plastic filter to have a place to fit, the handles must be cut out. These handles are created using the "Remove by rotation" and "Remove by extruding" commands. By rotation, we ensure that the semi-circular shape of the protrusion has nowhere to fit, and the extension is used to remove the material. First, one recess is formed at the top of the cover and again by mirroring it is formed at the bottom. Then we chose the view through the "cut" function for inspection.

There are four protrusions around the perimeter of the plastic filter, which means that it is necessary to create another two recesses on the sides. Again, only one was created and mirroring was used (see Figure 5).

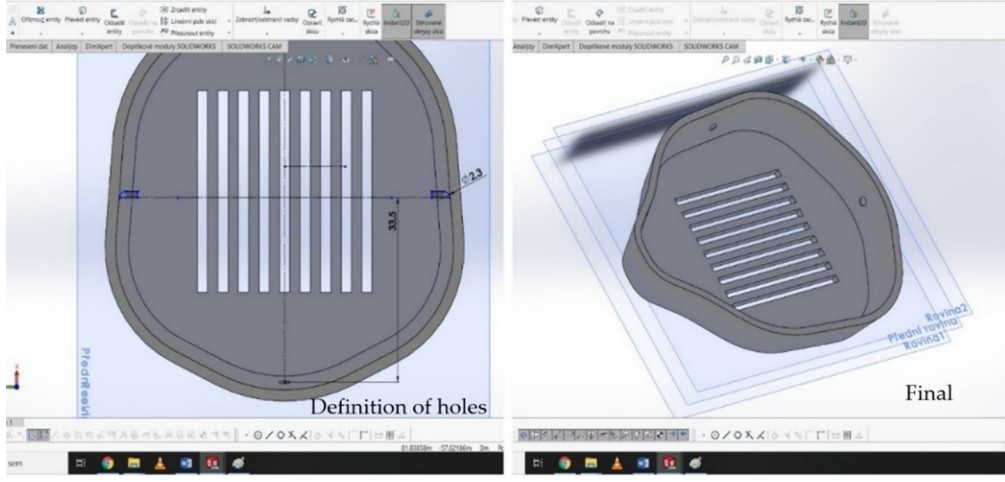

**Figure 5.** Mirroring holes and looking in planes (individual steps).

The only thing left is to model the body of the respirator. Most of the steps have already been repeated, and thus drawing a sketch with planes, then joining the profiles, cutting the semi-circular holes using the rotation function. The main task was to model the body of the respirator and create protrusions with holes on it so that it can be attached with, for example, a cord and a hole to a plastic and unprinted filter.

When creating a hole for a plastic filter, it started by drawing a hole in a 2D model. Its roundness and dimensions were determined. A view between the planes was chosen to check the correctness of the selection.

The selection of the material was continued so that the plastic filter had nowhere to fit. Again, the first drawing in 2D models was repeated. After removal, the view into the planes and the removal of the material were selected.

However, the semi-circular holes for the handles were still missing so that the plastic filter would not tend to fall out. A hole was created on the side, which was aligned exactly to the center and was created on the opposite side by mirroring. The process was repeated for the remaining two sides (see Figure 6). Subsequently, we switched to elevated body modelling.

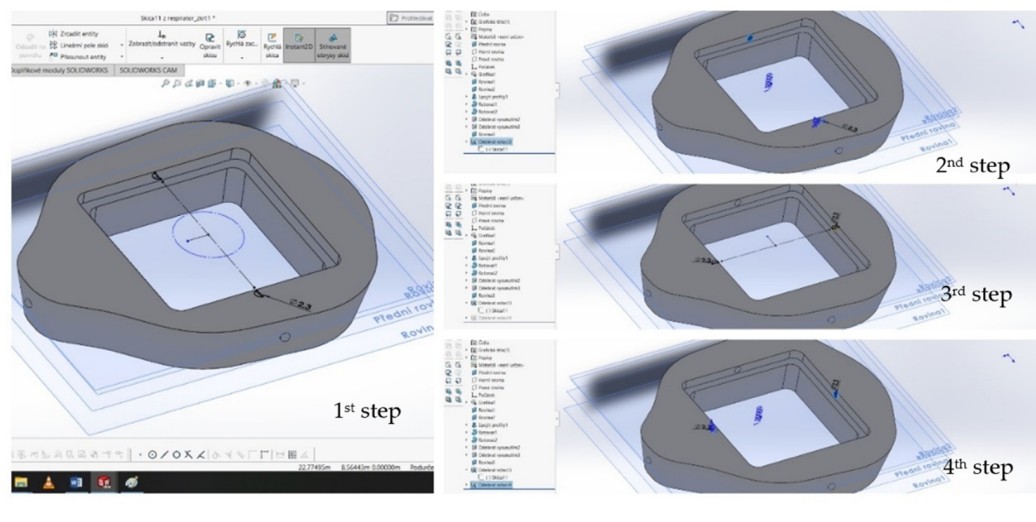

**Figure 6.** Creating semi-circular holes by selecting material (individual steps).

Skills in Solidworks is required to create a raised body. A downloaded model of the respirator was used, showing how it was modelled. First, a "goblet" was created, from which we gradually created the desired shape so that it would fit the human face. It consisted of up to seven planes if we did not count the front plane. This was followed by joining the profiles (thin walled). The model was projected back into 2D and a side view was used. When viewed from the side, it is more advantageous to model the selection of this type of material and the determination of the curve. In Figure 7 the same material selection is shown from the bottom and front of the respirator, respectively.

This was followed by the next part of creating the body, namely the creation of handles for attaching the respirator around the face. These were created by several functions such as material removal, material ejection, plane creation, profile joining, mirroring, and edge rounding. The entire process of forming the grips is shown in Figure 8.

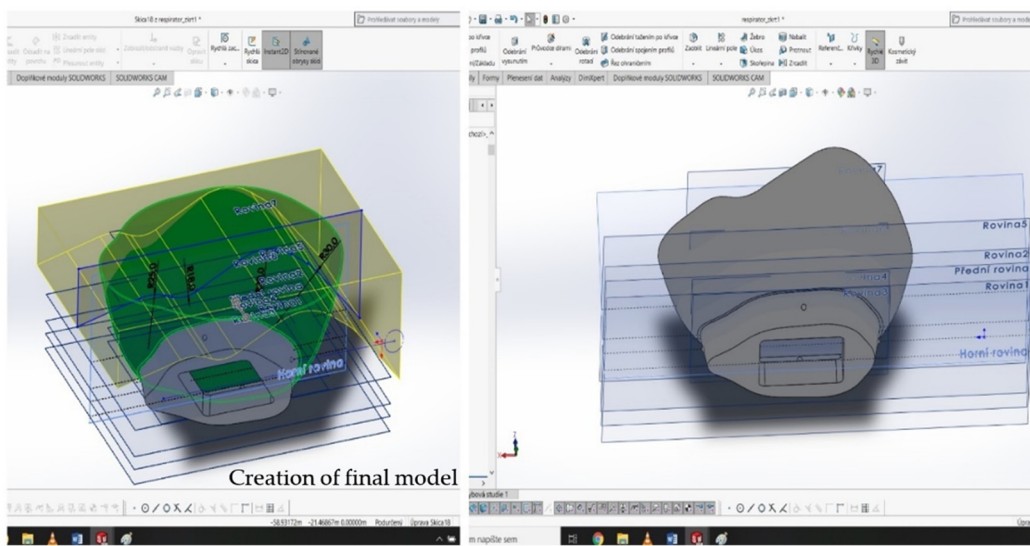

**Figure 7.** Carving the outside—view from below.

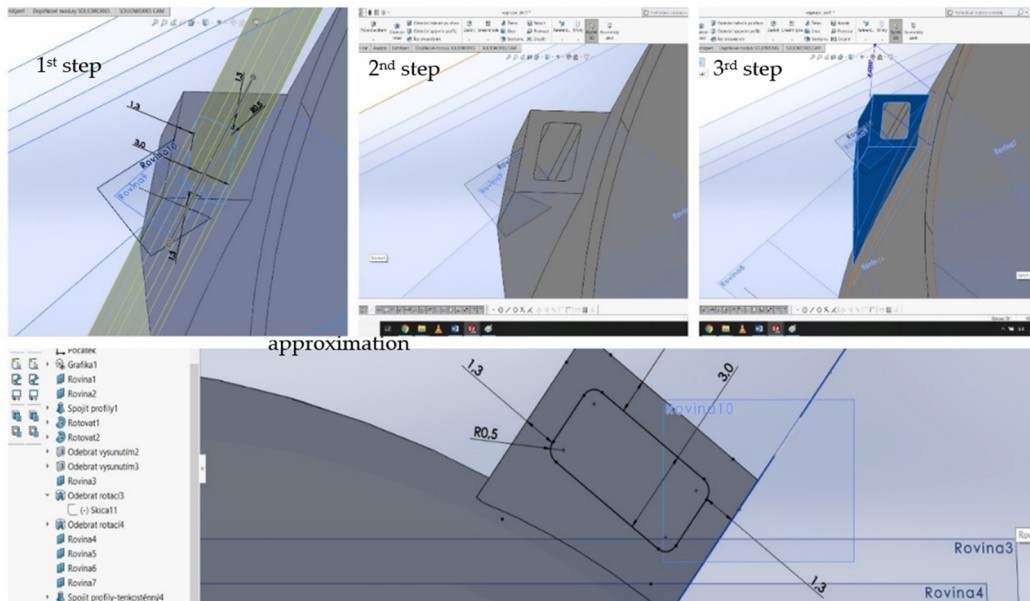

**Figure 8.** Creating a hole for the possibility of attachment, bonding profiles, and moving the hole.

## 3. Description of the 3D Printing Procedure of the Selected Object

For processing the design of a respirator suitable for the 3D printer, the material was selected and also the time of printing was determinate. Zmorph 3D printer was used to produce the model of the respirator. Dimensions, printing technology, and software was suited. The ZMorph printer solves complex solutions for 3D printing using several materials or machining directly on the desktop.

Advantages:

- Interchangeable tool heads;
- Can print thick paste as well (ceramics, dough, chocolate);
- CNC machining, engraving, two-color 3D printing;
- Print two materials simultaneously;
- Very high print quality;
- Handles high-level professional models;
- LCD display.

The ZMorph VX [17] features a rugged aluminium construction with high quality components that is designed to withstand high 3D printing speeds as well as high torque. Basic specifications of the 3D printer ZMorph VX is shown in Table 2.

**Table 2.** Basic specifications of the 3D printer ZMorph VX.

| **ZMorph VX** | |
| --- | --- |
| dimensions | $530 \times 555 \times 480$ mm |
| weight | 14 kg |
| software | Voxelizer 2.0.0 |
| print area size | $235 \times 250 \times 165$ mm |
| possibility of connection | USB, SD, Ethernet |
| printing on material | ABS, PLA, PET, ASA, HIPS, TPU |
| print about layer strength | 1.75–3 mm |
| price | 5374.82€ |

After much deliberation, it was decided to use ABS (acrylonitrile butadiene styrene) strings to make the respirator. Visible and tactile indistinguishable from PLA (polylactide) type strings. The entire respirator consists of three separately printed parts: 1. respirator body, 2. plastic filter, and 3. Filter cap. Each of these sections has been modified in Voxelizer 2.0.0. This program is recommended for the given printer and is therefore completely adapted for editing models that will be printed with the help of the Zmorph 3D printer, see Figure 9. It also offers editable settings for 2D and 3D milling, cutting, and laser engraving. It supports .dxf, .jpg, and .stl files. In Figure 8 we can see the body of the respirator designed in the program Voxelizer 2.0.0. Its position and dimensions have been adjusted so that we have enough material and so that printing does not take too long, as the Zmorph printer is not one of the fastest models of 3D printers. One such model of respirator body was extruded in approximately 80 min. The design of the entire respirator took about 180 min. After selecting the printer, the material and designing the model, a step follows, which consists of transferring the STL models to the software, where the appropriate G-codes have been read and created. After adding G-codes, there were two options for data transfer, either directly to the printer or using an SD card. Specifically, for the Zmorph printer, the SD card method was chosen. This process is no different for any type of model.

The second part of the respirator model is a plastic filter. It had to be adapted in size and shape to the other two parts of the respirator so that it would fit exactly in the specified position. The task, after being placed in the body of the respirator, was to filter the air. The plastic filter went through the same process as the body of the respirator. It was designed and modified in the same software Voxelizer 2.0.0. Voxelizer also calculated the expected printing time of the two plastic filters, which was 1 h and 56 min. An alternative to printing two pieces has been chosen so that the printing area and printer performance can be used to the maximum. In the program, it is possible to adjust the position where the printout will be printed, the thickness of the printout, etc. The third and last printed part of the respirator model is the filter cover. Its task is to prevent damage (e.g., to a mechanical) unpressed filter and at the same time forms the front part of the entire respirator. A green ABS string was used to make this part of the respirator. The model also underwent thorough preparation for printing in the Voxelizer program.

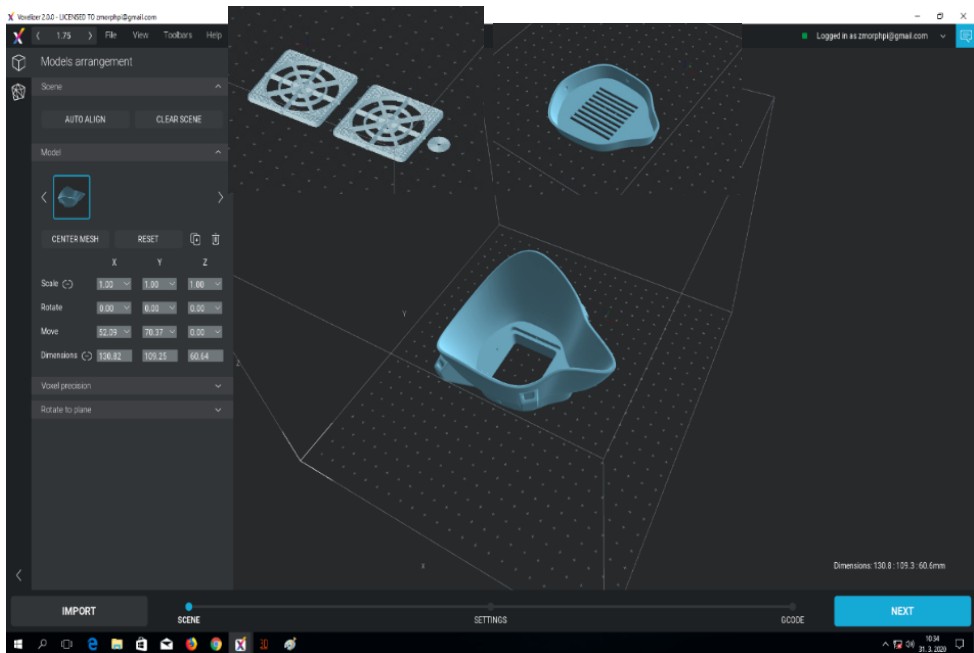

**Figure 9.** Respirator body, two pieces of plastic filter and plastic filter cover in Voxelizer software.

*Print a Respirator on a ZMorph Printer*

The ABS/Acrylonitrile butadiene styrene was chosen to produce the respirator. Visible and unrecognizable from PLA/polylactide strings, ABS plastic has been used in automotive manufacturing for years, as it demonstrates resistance to oils, acids, and moisture. Its thermal stability is satisfactory. It does not have an exact melting point; it is around 180 °C. The disadvantage is the relatively high temperature for solidification of the material, which is about 100 °C. There is a high risk of material loss during the cooling process. This can result in a loss of up to 0.8% of the total volume. As a result, significant deformations of the model (e.g., twisting of the first layers) can occur until the model itself cracks. This problem can be solved by choosing a printer. This is ideal if the printer has a heated work platform. The ideal temperature of this pad is between 100 and 130 °C. You also need to focus on the printing process itself. Caution is required when heating the plastic, releasing acrylonitrile (a toxic compound) fumes, which can result in severe mucosal irritation. Due to the ability of ABS to have good solubility in acetone, it is possible to produce large models and then glue them. After selecting the printer, material, and model design, it is necessary to transfer the STL models to the software and create the appropriate G-codes. After adding G-codes, there were two options for data transfer, either directly to the printer or using an SD card. Specifically, for the Zmorph VX printer, the SD card method was chosen, see Figure 10. This process is no different for any type of model.

The first model was a printed respirator using a ZMorph printer. We started by printing the second part of the model and thus the plastic filter. Two pieces of plastic filter software calculated for 1 h and 56 min of printing. This piece lasted 60 min. The ZMorph printer is not characterized by fast printing, but the quality of the prints is at a high level. The whole printing of the two pieces took the expected 1 h and 56 min. The "propeller" is sufficiently reworked. No major deficiencies or excess material are detectable. As a second model, the body of the respirator was made. It is not much more difficult to print and was printed only in one piece. Significantly more material was used, as the model does not contain as many holes as a plastic filter. It contains only one large hole in the middle and four smaller holes on the sides. In Figure 11 is the body of the respirator pressed from a red ABS string. The rolls of strings are placed above the printing surface to ensure the supply of material in an uncomplicated form. It took about 85 min to press the body of the respirator. The extruded heat also has no major drawbacks. The dimensions remained

intact. Only the sequence of application of the last layers can be seen. However, this does not affect the quality and function of the printout.

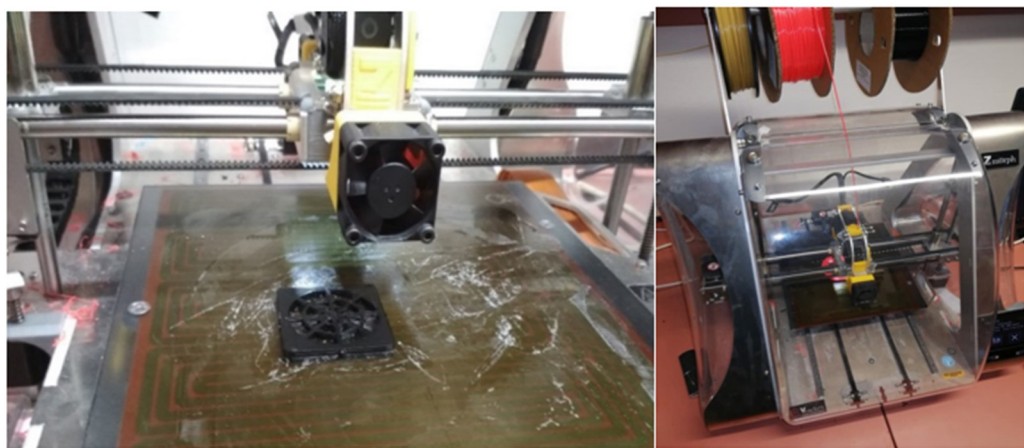

**Figure 10.** A more detailed shot of 3D printing on the ZMorph printer.

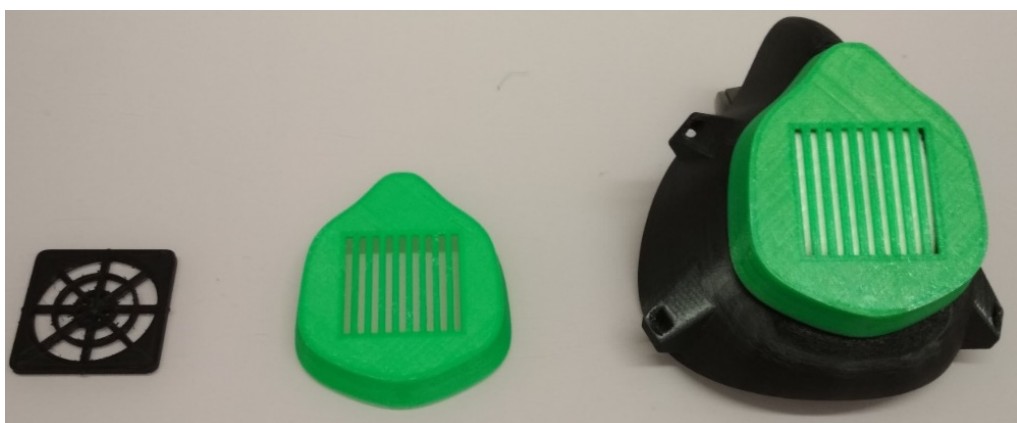

**Figure 11.** Complete respirator and its separate parts.

Subtle imperfections can be observed in the individual parts. A closer look at the respirator shows the stratification of the string with the naked eye. With the touch of a finger, it is possible to feel traces of the string or the transitions from layer to layer. When making holes to hold the respirator around the face, the ZMorph printer hesitated slightly, However, it does not affect the functionality of these holes. It is also possible to see transitions between the layers on the folds of the filter cover. Darker places are visible mainly on the main connections. When looking at the respirator from the top, the X-shaped layers can be seen on the filter cover. The shape of each gap is the same and there are no differences in the layers of the individual grids. As well as on the sides and at the top, there are darker places. With an unprinted filter inserted, the respirator provides full protection. It is only necessary to use the material to be tied around the head. However, the respirator is not certified and therefore cannot be used by paramedics and front-line people. The ZMorph [17] printer confirmed the expected print quality. Printing one complete respirator takes about 3 h and 40 min. The complete respirator and its separate parts its show in Figure 11. It is possible to adjust the size of the respirator according to the size of the shape—a man or a woman to fit exactly on the face. Only one size was printed in the case study. As the PLA material deforms, the ABS material was chosen because it does not deform outdoors or under the influence of sunlight. However, for indoor use, it does not matter.

Models printed with the ZMorph VX printer are flawless and of high quality, even when printing much more demanding models. Although the time required for printing is

longer than with the TriLab DeltiQ XL printer, the resulting product has a significantly better processing quality. The main advantage of the ZMorph VX printer is its multifunctionality. Whether hybrid printing (printing two different materials at once), the possibility of CNC machining or engraving.

Specific parameters of respirator:

- Washable and disinfectable respirator with replaceable filter;
- Different materials can be used as a filter—the recommendation for achieving high quality is colloidal silver;
- The inner space of the filter can be modified;
- The respirator has an ergonomic shape for most face types—resize by adjusting the STL model;
- The cord is not part of the 3D printing—it must be rubber, resp. another improvised means.

Table 3, [18] shows the processed selected parameters in comparison to the different types of respirators to prove the validity of the implementation of the respirator's own design.

**Table 3.** Selected parameters for respirator comparison.

| Respirator | Selected Parameters | |
| --- | --- | --- |
| | **Filter** | **Ergonomic Shape** |
|  | replaceable colloidal silver | most face types |
|  | replaceable filters of different materials | most face types |
|  | replaceable cotton or gauze | most face types |

**Table 3.** *Cont.*

| Respirator | Selected Parameters | |
|---|---|---|
| | **Filter** | **Ergonomic Shape** |
|  | replaceable filters of different materials | two models: suitable for a round face and oval face |
|  | replaceable homemade filters of different materials | most face types |
|  | Cotton | most face types |

## 4. Conclusions

Initially, 3D printers were used by larger corporations to produce prototypes, which were then used to modify the final version of their product to meet requirements. 3D technology is not universally suitable for various types of products. It is only a matter of time before 3D printing produces larger volumes. The advantage of using plastics in 3D printing is the possibility of re-recycling, which is, however, limited by the number of recycles, as plastics become contaminated in some way and when they melt, their properties are degraded. The material can go through five cycles if it becomes unusable. Nevertheless, it can still be used in others, perhaps to produce such precise objects or products—it can still be used as a material [8,19,20]. Another interesting material is metal waste, which is sorted, cleaned, crushed to dust, and can be used for 3D printing of metal parts. Of interest is the use of oil used in the preparation of food in McDonald's, where the aim is to process it into a photopolymer resin, which is suitable as a material for 3D printing [21–24]. There is a need to develop a more diverse biocompatible material with specific biological, mechanical, and chemical properties that can be used for the application of 3D (or 4D) bioprinting. Furthermore, it is necessary to develop 3D technologies with higher speed and print resolution. The production of 3D materials with living elements requires innovative studies that can be revived to introduce new features. The physico-chemical and mechanical properties of 3D printed materials must be further enhanced by changing the chemistry of the photopolymerization or by using additives to produce materials with the desired properties. At the beginning of the 21st century, the potential and use of 3D printing intensified in the field of medicine. Attempts can be made to produce a functional replacement organ in the body. Today, this potential is at a higher level and the first successful operations with organs produced with the help of 3D printing technology can be seen. It is possible to use 3D printing even during the production of small jewellery. The possibility of producing cars, food, or buildings using this method also comes to mind. In the USA, they started to use this method to produce equipment components for extreme conditions. These are spare parts for the repair of devices for units that are far from civilization. The growing popularity of 3D/4D printing is because in a relatively short time it is possible to produce a wide range of products from various materials and that are

intelligent for various industries [25–29]. In 2015, Joseph DeSimone introduced a technique to accelerate 3D printing with Carbon 3d at the University of North Carolina at Chapel Hill. This is an excellent material for 3D printing of prototypes and the production of real and functional parts. It has excellent mechanical properties, high resolution, and exceptional surface quality. Another goal to increase the quality of 3D printed products is to capture errors using a system of sensors and high-speed cameras, which are aimed at monitoring irregularities and then adjust them in real time. The importance of laboratories in research and development is considerable, especially in testing the possibilities of how to improve and enhance 3D printing and in developing a range of materials suitable for 3D printing, as well as exploring its potential in connection with virtual reality [30].

**Author Contributions:** Conceptualization, P.T. and M.P.; methodology, M.P. and S.K.; software, M.P. and M.K.; validation, P.T. and S.K.; formal analysis, P.T.; investigation, M.P. and M.K.; resources, M.P. and M.K.; data curation, M.P. and M.K.; writing—original draft preparation, M.P., P.T., M.K. and S.K.; writing— review and editing, M.P.; visualization, M.K.; supervision, P.T.; project administration, P.T. and M.P.; funding acquisition, P.T. and M.P. All authors have read and agreed to the published version of the manuscript.

**Funding:** This research received no external funding.

**Acknowledgments:** This article was created by the implementation of the grant project APVV-17-0258 "Digital engineering elements application in innovation and optimization of production flows", APVV-19-0418 "Intelligent solutions to enhance business innovation capability in the process of transforming them into smart businesses", VEGA 1/0438/20 "Interaction of digital technologies to support software and hardware communication of the advanced production system platform", KEGA 001TUKE-4/2020 "Modernizing Industrial Engineering education to Develop Existing Training Program Skills in a Specialized Laboratory".

**Conflicts of Interest:** The authors declare no conflict of interest.

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
