# Peer review of "Case Study: 3D Modelling and Printing of a Plastic Respirator in Laboratory Conditions"

_applsci, doi:10.3390/app12010096_

Round 1

Reviewer 1 Report

For those who are not familiar with this working method of 3D printing, the authors make a very correct and didactic presentation of the method. From this point of view, I agree with its publication without any supplementary request.

From the point of view of the purely scientific contribution - it shows the way in which the software is used, and the utility of using 3D printing in all aspects. As a result, only the introduction of comparative and statistical data adds some scientific value.

Author Response

Dear Reviewer 1, Thank you very much for your Review. The article will be modified according to the comments in your review. Does the introduction provide sufficient background and include all relevant references? The introduction was supplemented to increase the quality from the References processed information. Best regards. Authors

Reviewer 2 Report

This paper presents a 3D printing process for manufacturing a respirator. It covers the following topics: (1) 3D printing, (2) block-chain, and (3) the modelling and manufacturing procedures of a respirator. The follows are the comments toward this article:

(1) The case study is not scientific or engineering significant. The target model is not difficult to make by using additive manufacturing technology.

(2) Section 2 and 3 should be shortened. The modelling and manufacturing processes possess no technical merit. 

(3) In the introduction section, 3D printing and block-chain are discussed in parallel. This is a very good feature and deserves more exploitation. At current moment, the transportation in the whole world is under restriction because of COVID-19. 

(4) The authors may spend efforts to distribute or to upload the G-code programs into the internet by using blockchain platforms and perform an emulation to demonstrate how blockchain can achieve in improving the security, usability, and management of 3D printing digital intellectual properties, including G-codes and geometric models.

Author Response

Dear Reviewer 2, Thank you very much for your Review. The article will be modified according to the comments in your review. The article was expanded. (1) The case study is not scientific or engineering significant. The target model is not difficult to make by using additive manufacturing technology.

The aim of the article was not only to produce a respirator, but rather to design it and set up printing in a laboratory conditions. It was the first prototype that was printed in the laboratory conditions at our workplace.

(2) Section 2 and 3 should be shortened. The modelling and manufacturing processes possess no technical merit.

For the reason set out in point (1), we consider that these procedures needed to be described in more detail.

(3) In the introduction section, 3D printing and block-chain are discussed in parallel. This is a very good feature and deserves more exploitation. At current moment, the transportation in the whole world is under restriction because of COVID-19.

As authors, we see great importance and potential in the interconnection of 3D printing and Blockchain, not only in the context of Covid-19, but also in the context of digitization of the value chain. For this reason, we dedicate a certain part of the article to this issue. However, as a workplace, we do not yet have the opportunity to try this technology.

(4) The authors may spend efforts to distribute or to upload the G-code programs into the internet by using blockchain platforms and perform an emulation to demonstrate how blockchain can achieve in improving the security, usability, and management of 3D printing digital intellectual properties, including G-codes and geometric models.

The use of G-codes has its great rationale for 3D printing and in the link to Blockchain. A number of professional papers have been published on this topic, some of which are also listed in the References. Best regards. Authors

Reviewer 3 Report

The manuscript presents a 3D printing technology for a plastic respirator. However, this manuscript doesn't look like a scientific paper. In fact, 3D printing is not new technology. Authors used FDM technology with popular materials without new approaches. For this product, there are some other methods that are better than 3D printing, such as molding technology. 3D printing takes a long fabricating time, and the surface is rough. There are some issues that should be addressed:
- Please check the English style.
- What are the effects of Blockchain on your model and product? Why does blockchain only mention in the Introduction?
- What are the characteristics of the used filament? Printing direction? The temperature of the nozzle and base? 
- For the completed product, it should be compared with other products, as well as evaluate usability in a real environment. Lack of detailed evaluations of the properties of the product.
- For pictures, please prepare carefully: add sub-names for each sub-picture and remove the window taskbar.
- For References, this section is not enough for the paper. Besides, authors should limit the websites that show as a reference.
Totally, this manuscript does not include enough new contributions and necessary analyses.

Author Response

Dear Reviewer 3,

Thank you very much for your Review. The article will be modified according to the comments in your review. The article was expanded. The changes were focused on comments.

  • Please check the English style.
  •  
  • Was revised.

  • - What are the effects of Blockchain on your model and product? Why does blockchain only mention in the Introduction?
  •  
  • The meaning of Blockchain has been processed in general, as we see in it the importance of its use. However, as a workplace, we do not yet have the opportunity to try this technology. The impact and relationship to our product is evident in the general approach. As authors, we see great importance and potential in the interconnection of 3D printing and Blockchain, not only in the context of Covid-19, but also in the context of digitization of the value chain.
  •  
  • - What are the characteristics of the used filament? Printing direction? The temperature of the nozzle and base?
  •  
  • Has been expanded.
  •  
  •  For the completed product, it should be compared with other products, as well as evaluate usability in a real environment. Lack of detailed evaluations of the properties of the product.
  •  
  • Has been expanded in article.
  •  
  • - For pictures, please prepare carefully: add sub-names for each sub-picture and remove the window taskbar.
  •  
  • Was corrected.
  •  
  • - For References, this section is not enough for the paper. Besides, authors should limit the websites that show as a reference.
  •  
  • Has been expanded.
  •  
  •  
  • Totally, this manuscript does not include enough new contributions and necessary analyses.
  •  
  • As authors, we believe that the significance of the article is in practical application in laboratory conditions at the time of the Covid-19 pandemic in its initial phase, when there was a shortage of protective equipment in our country. In this way, assistance to critical infrastructure was offered.
  •  
  • Best regards. Authors

Round 2

Reviewer 2 Report

  1. In the introduction, the importance of block-chain had been clearly presented.
  2. In the test section (Sec. 3), the differences between the thermal plastic filaments  had been addressed.
  3. However, one more suggestion, why not upload the digital contents (the geometric models and the G-codes) to a block-chain platform and show the encoded data as well as the reliance  and security provided by blockchain technologies ? 

Author Response

Thank you very much for your review. The article will be modified according to the comments in your review. The article was expanded.

This will be an effort in further research.

Best regards.

Authors.

Reviewer 3 Report

What are the effects of the blockchain on your model and product? This content has been done in the Introduction section, which is not enough. If no experiments, it is just a discussion or an idea and should be added to the Discussion section.
Besides, please remove the Window taskbar in the figures.
Lack of detailed evaluations of the properties of the product when used in a real environment.
Please compare the product with other studies to highlight the contributions of the manuscript.
I don't think that the current version of this work is enough to publish. Please add more analysis and more contributions. 

Author Response

Thank you very much for your Review. The article will be modified according to the comments in your review. The article was expanded.

As authors, we see great importance and potential in the interconnection of 3D printing and Blockchain, not only in the context of Covid-19, but also in the context of digitization of the value chain. For this reason, we dedicate a certain part of the article to this issue. However, as a workplace, we do not yet have the opportunity to try this technology.

The window taskbar in the figures was removed.

Evaluation of the properties of the product has been added on page 11.

The comparison of the product is Tab. 3.

Best regards.

Authors.

Round 3

Reviewer 3 Report

Thanks for the revision.